# Translation and Cultural Adaptation of the Patient Self-Administered Financial Effects (P-SAFE) Questionnaire to Assess the Financial Burden of Cancer in French-Speaking Patients

**DOI:** 10.3390/healthcare8040366

**Published:** 2020-09-25

**Authors:** Dominique Tremblay, Thomas G. Poder, Helen-Maria Vasiliadis, Nassera Touati, Béatrice Fortin, Lise Lévesque, Christopher Longo

**Affiliations:** 1Centre de Recherche Charles-Le Moyne–Saguenay–Lac-Saint-Jean sur les Innovations en Santé, 150 Place Charles-Le Moyne, Longueuil, QC J4K 0A8, Canada; helen-maria.vasiliadis@usherbrooke.ca (H.-M.V.); beatrice.fortin@usherbrooke.ca (B.F.); lise.levesque@usherbrooke.ca (L.L.); 2Faculté de Médecine et des Sciences de la Santé, Université de Sherbrooke, Campus de Longueuil, Longueuil, QC J4K 0A8, Canada; 3Département de Gestion, Évaluation et Politique de Santé, École de Santé Publique de l’Université de Montréal (ESPUM), Université de Montréal, Montréal, QC H3N 1X9, Canada; thomas.poder@umontreal.ca; 4Centre de Recherche de l’Institut Universitaire en Santé Mentale de Montréal, CIUSSS de l’Est de l’Île de Montréal, Montréal, QC H1N 3M5, Canada; 5École Nationale D’administration Publique, Montréal, QC H2T 2C8, Canada; nassera.touati@enap.ca; 6DeGroote School of Business, McMaster University, Hamilton, ON L8S 4L8, Canada; cjlongo@mcmaster.ca

**Keywords:** financial burden, questionnaire, Quebec, French, patient-reported experience, cancer, translation

## Abstract

People living with and beyond cancer (PLC) experience financial hardship associated with the disease and its treatment. Research demonstrates that the “economic toxicity” of cancer can cause distress and impair well-being, health-related quality of life and, ultimately, survival. The Patient Self-Administered Financial Effects (P-SAFE) questionnaire was created in Canada and tested in English. The objective of this study is to describe the processes of translation and cultural adaptation of the P-SAFE for use with French speaking PLC in Canada. The Canadian P-SAFE questionnaire was translated from English to French in collaboration with the developer of the initial version, according to the 12-step process recommended by the Patient-Reported Outcome (PRO) Consortium. These steps include forward and backward translation, a multidisciplinary expert committee, and cross-cultural validation using think-aloud, probing techniques, and clarity scoring during cognitive interviewing. Translation and validation of the P-SAFE questionnaire were performed without major difficulties. Minor changes were made to better fit with the vocabulary used in the public healthcare system in Quebec. The mean score for clarity of questions was 6.4 out of a possible 7 (totally clear) Cognitive interviewing revealed that lengthy questionnaire instructions could be confusing. Our team produced a Canadian-French version of the P-SAFE. After minor rewording in the instructions, the P-SAFE questionnaire appears culturally appropriate for use with French-speaking PLC in Canada. Further testing of the French version will require evaluation of psychometric properties of validity and reliability.

## 1. Introduction

Major advances in cancer diagnosis and treatment have led to an increase in the number of people living with and beyond cancer (PLC) [1,2,3]. Cancer is experienced as a complex disease that impacts all aspects of life [4,5]. Along with physical and emotional challenges, PLC face financial burdens [6,7,8] related to out-of-pocket costs (OOPC). Multiple factors contribute to the burden of OOPC, such as a decline in income due to difficulties remaining at or returning to work after cancer diagnosis [9]. Loss of income often coincides with new expenses: modifications to the home, special clothing, exercise equipment or programs, altered diets, drug costs, child care, home care, transportation costs [10]. Along with their own decline in income and rise in expenses, PLC worry about their caregiver’s lost wages from days off to care for them and accompany them to medical treatments and outpatient consultations. Government programs or private insurance may moderate these financial burdens, but limited access or insufficient coverage decreases the effectiveness of this safety net for a number of PLC and caregivers [11].

Recent research strongly suggests that the financial burden of cancer impairs well-being, health-related quality of life, and ultimately survival [12,13,14]. A growing body of literature from North America and Europe reveals that the financial burden is experienced by PLC [15,16,17] even in universal healthcare systems [10,18,19,20]. In Canada, some PLC suffer bankruptcy, loss of the family home, and high levels of distress [11]. Coping strategies used by PLC facing financial hardship include non-adherence to therapy, skipped medical visits and foregoing beneficial lifestyle habits [6,21,22,23,24]. PLC with a heavier financial burden have lower adherence to cancer treatments, shorter survival, poorer prognosis, greater risk of recurrence [24], more physical symptoms, more severe anxiety and depression, and poorer perceived health status [25]. Together, these effects are referred to as the “financial toxicity of cancer” [24], which is recognized as a clinically relevant adverse patient-reported outcome (PRO). PLC also report unmet needs for information pertaining to financial issues such as insurance coverage and the availability of services and support [26].

Studies suggest that, in publicly funded health systems such as Quebec’s, PLC who are under 65 years old, are unemployed, or have lower levels of education, are at greater risk of facing financial hardship [7,25,27,28]. Efforts have been invested in better understanding the coping strategies of PLC and their impact on the cancer trajectory [10,18,29,30]. However, the nature and extent of the financial burden remain understudied and inadequately addressed.

The financial burden is closely linked to a country’s socioeconomic context and health system [24]. It is therefore important when seeking to assess the financial burden of cancer to select measurement instruments that take these context elements into account. In Canada, Christopher Longo, University of McMaster (Canada) has undertaken pioneering work in evaluating the perceived financial burden faced by PLC. He developed the Patient-Self-Administered Financial Effects (P-SAFE) questionnaire, which focuses on the financial impact of cancer [31]. This instrument includes 23 questions: 16 are multiple-choice questions using various scales, with or without fields to add information; seven involve written responses in various formats, including completing tables. The questionnaire begins with detailed instructions. P-SAFE has been tested for content validity in the Canadian context and a growing number of studies have reported results generated with this instrument. P-SAFE was first used in a pan-Canadian study [32], that excluded the province of Quebec because there was no French version of the Canadian questionnaire. Statistics Canada data from 2016 show that 79.1% of people living in Quebec and 2.2% of people living in other parts of Canada speak only or mostly French at home [33]. The lack of a French-language version of the questionnaire impedes the assessment of the global financial burden of cancer across Canada, as well as a comparison between provinces. With almost 8.5 million inhabitants, Quebec represents 22.5% of the Canadian population.

Quebec has a publicly funded healthcare system with universal coverage. However, provincial programs do not cover all expenses. The financial burden may vary between PLC according to the type of cancer and treatment, as well as factors such as complimentary private insurance coverage for some cancer drugs; family support; employment situation; and distance from cancer centre. A better understanding of the financial burden of cancer is needed to improve decision-making at multiple levels, from government policy around coverage, to health services organization, employment practices, treatment choices and more [34]. Interestingly, a recent study found that PLC did not mention financial distress to their healthcare professionals [35]. In publicly funded health systems, many people, including healthcare providers, believe expenses for serious diseases are covered, and may be less aware and less equipped to address financial questions [8,36]. The need to raise awareness of and attention to the growing financial burden of cancer motivated the effort, described in this paper, to translate and adapt the P-SAFE questionnaire for French-speaking PLC in Quebec.

This paper presents the process used to produce a culturally adapted French-language version of the P-SAFE questionnaire, a process that incorporated PLC perspectives. This initial testing phase is necessary to provide insight into how respondents understand and interpret the items and instructions on the translated questionnaire. The process aims to ensure consistency in the content and face validity between the English and French versions of the P-SAFE [37]. Content validity rests on qualitative, not quantitative, evaluation [38].

## 2. Materials and Methods

Translating the P-SAFE questionnaire involved both language and cultural considerations, with words, expressions, and items adjusted in order to capture the concepts from the original version in a way that would reach PLC in Quebec. The 12-step translation process of the Patient-Reported Outcome Consortium was followed to ensure that the integrity of the original measurement tool was maintained while being adapted for use in Quebec (Figure 1) [39].

### 2.1. Steps in the Process of Translation and Cultural Adaptation

#### 2.1.1. Steps 1 to 3: Preparation, Forward Translation and Reconciliation

In January 2019, our team contacted the developer of the original instrument to express our interest in translating the P-SAFE questionnaire for use with French-speaking PLC in Quebec, and enlist his collaboration. Forward translations were undertaken by two professional bilingual translators, whose mother tongue was Quebec French, working independently from each other [40]. This step produced French versions A and B of the P-SAFE questionnaire. The two versions were reviewed and compared to check for adequacy of vocabulary, grammar and comprehensibility. In Step 3, discrepancies between the versions were discussed until consensus was reached on the wording of questions and response choices. This led to a reconciled forward translation.

#### 2.1.2. Step 4: Back-Translation

The reconciled French language questionnaire was then translated back into English (versions C and D) by two professional translators whose mother tongue was Canadian English, working independently of each other; they remained blind to the original English P-SAFE questionnaire. The aim of back-translation is to highlight discrepancies with the original version, as well as assure cultural adaptation and help generate consensus [37]. Our multidisciplinary expert committee reviewed the back-translations, as well as the differences observed between the two forward translations, and resolved any concerns to produce a final consensus version.

#### 2.1.3. Steps 5 to 7: Review of All Versions, Adaptation, Harmonization and Proofreading

For Steps 5 to 7, the authors set up a multidisciplinary expert committee including a nurse researcher with expertise in cancer services research (DT) and a health economist (TGP), both of whom had prior experience with the translation of measurement instruments [41,42]. The committee also included a public health researcher with expertise in cost–benefit analysis (HMV) [43], a researcher with expertise in organizations and governance (NT) [44], and a bilingual (French and English) research assistant (LL). The committee reviewed the four versions (A, B, C, D) and produced a harmonized French language experimental Canadian P-SAFE questionnaire. Expert committee members made decisions by consensus during proofreading around the equivalence of French and English questionnaires, guided by the following questions: (a) Are the domains conceptually equivalent (relevance, meaning, and importance) in each language? (b) Are items equally relevant and acceptable in each culture (item equivalence)? (c) Is the meaning of the item the same in each culture (semantic equivalence)? (d) Can the questionnaire be used in the same way by PLC (operational equivalence)? Expert committee comments on the questionnaire are summarized in Table 1. In the course of these discussions, one item was added to the questionnaire to address concerns among members of the committee that patients faced with financial hardship were more likely to discontinue or interrupt treatment; this was something they observed in practice and was supported by findings of a systematic review [43]. Finally, a synthesized French version was produced and discussed with Professor Longo.

#### 2.1.4. Steps 8 to 10: Pre-Test, Cognitive Interviewing and Post-Cognitive Interviewing

The purpose of pretesting the experimental French P-SAFE questionnaire was to determine the level of clarity and cultural compatibility of the self-administered questionnaire from the point of view of PLC.

### 2.2. Setting, Recruitment and Participants

A purposive sample of adults receiving ambulatory cancer care was recruited between May and August 2019. The eligibility criteria were: ability to read and understand French, and having completed an initial cycle of treatment for breast, colorectal, prostate or lung cancer—the most common cancers [45]. Ambulatory patients having completed at least a first cycle of treatment were selected as they were sure to have been exposed to cost burdens. Participants were informed about the study by a member of their care team (usually a social worker or nurse). Interested PLC were referred to the research assistant, who explained the study and obtained informed consent from the participant. Of the eight individuals who contacted the assistant, seven completed the interview.

### 2.3. Procedure

PLC completed the French P-SAFE questionnaire in the presence of a research assistant. They were asked to assess the clarity of each question on a scale of one to seven (range 1 = not at all clear to 7 = totally clear) [38] and describe what they were thinking and feeling as they completed the questionnaire [46]. The research assistant then conducted a debriefing interview to elicit opinions or emotional reactions, ask about any unclear items or expressions used in the questionnaire and any issues around choices of response. Participants were invited to contribute their perspective on: (1) what might facilitate use of the tool; (2) potential barriers to use; and (3) ways to optimize acceptability. Participants also completed a socio-demographic questionnaire. The study was approved by the Research Ethics Board of the Centre intégré de santé et services sociaux de la Montérégie-Centre (MP-04-2019-316). Informed written consent was obtained from all participants.

### 2.4. Data Analysis 

Descriptive statistics were performed on the clarity of each question, based on the numerical rating scale [38]. A single table was used to categorize the think-aloud and debriefing data. This table organized all the issues identified by PLC participants alongside the decisions of the expert committee regarding adapted equivalence dimensions. The analysis was led jointly by DT and LL, who read, coded, and synthesized the data to meet the study objectives, refining the process through discussion with the multidisciplinary expert committee, and achieving consensus on interpretations.

## 3. Results

Seven PLC participated in the pre-test and interview (Table 2). This is a satisfactory sample size for pretesting an experimental translated version of a questionnaire [37].

### Perspective of People Living with and beyond Cancer (PLC)

From the PLC perspective, the number of questions (*n* = 23) was acceptable, and the questions were deemed to be simple and not intrusive. Participants took approximately 20 min to complete the questionnaire. Scores for the clarity of each question ranged from 5 to 7 (mean 6.4) out of 7. PLC questioned the meaning of instructions for question 4 and, to a lesser extent, for questions 1 and 5. They reported that the instructions were so detailed they could be confusing. (Of note, we decided to keep the instructions as they were and evaluate the impact on non-response rates when administering the questionnaire to a larger sample). The items were considered culturally well adapted and the response choices appeared logical. This stage of testing confirmed that the French version of the P-SAFE questionnaire was functionally equivalent to the original English version for use in the Canadian publicly funded healthcare system. 

Changes were made to questionnaire instructions, questions and response choices to address issues raised by the expert committee. As seen in Table 1, several changes were proposed to address concerns about the appropriateness of certain phrasings. For example, the expression “your cancer” was replaced by “a cancer”, or just “cancer”; and the negative phrasing “lesser value housing” was replaced by the positive “affordable housing”; the word “patient” was replaced by “you”. Other comments reflected concerns with specific terminology (i.e., related to insurance) that would require further verification. Still, others addressed awkward differences in the phrasing of questions (these were standardized). Concerns that the choice of income categories be comparable with national survey instruments led to harmonization with Statistics Canada income categories. Lastly, as noted above, one question was added to the questionnaire to capture the impact of financial hardship on adherence to treatment. 

## 4. Discussion

This study describes a systematic standardized process for the translation and cross-cultural adaptation of the P-SAFE questionnaire for use with French-speaking PLC in Quebec. The PLC who participated, the multidisciplinary expert committee and the P-SAFE developer collectively represent a well-informed group for this effort. The process generated a translated version with satisfactory semantic equivalence to the original that was clear and culturally acceptable [46]. According to Vallerand [38], a clarity score above four suggests that a questionnaire item does not require revision.

The French-language P-SAFE questionnaire uses the same format and, with only a few adjustments, the same instructions as the original version, which is fundamental for an instrument to produce comparable results [38]. PLC participants in the pre-test interview agreed that the P-SAFE questionnaire was relevant as a detection tool to better understand the financial burden of cancer. It was viewed as a promising way to increase attention to and follow-up of financial distress. Results suggest that the questionnaire may facilitate communication between PLC and cancer care providers.

### 4.1. Strengths and Limitations

The study followed guidelines [39] for the cross-cultural adaptation of the P-SAFE questionnaire. This approach increases the probability of equivalence between the French version and the original English version in terms of item, semantic, operational, measurement, and functional equivalence, and allows greater comparability of responses across populations. The back-translation method represents a particular strength of the study, since P-SAFE developers are not familiar with the French language of Quebec. The approach also reduces the potential for bias related to personal characteristics, linguistics and comprehension.

The multidisciplinary expert committee played a crucial role in reviewing all versions, making key decisions, reaching consensus on discrepancies, and consolidating all versions of the survey [46]. Professional translators and committee members with relevant experience were selected according to cross-cultural adaptation guidelines [38]. The “think-aloud” approach, which encouraged PLC to voice their thoughts while answering the questionnaire, followed by cognitive interviewing, highlighted issues around the clarity of instructions that might otherwise have been overlooked. 

Limitations of the study should be mentioned. The study population from Quebec may not be representative of other French-speaking people in other jurisdictions. It can therefore not be assumed that the questionnaire is suitable for use in other countries where French is spoken: linguistic, cultural and health system characteristics may be different from those present in Quebec (and there may be slight differences even in other Canadian provinces). Another limitation relates to the sampling strategy and demographics of PLC participants in the pre-test and interview. A convenience sample was employed due to the challenge of recruiting PLC who had completed an initial cycle of treatment. 

### 4.2. Clinical and Research Implications

The scientific literature provides recommendations for cross-cultural adaptation of questionnaires [46]. These authors warn that translating and adapting a questionnaire for different cultural groups are two distinct processes requiring time, skill, experience and financial resources. Our study builds on existing studies conducted with the P-SAFE questionnaire, and our results suggest there would be benefits from using a broader sample in analyzing content validity. For the tool to be used in practice, it should ideally be studied for sensitivity to change in order to reflect challenges PLC experience across the cancer trajectory [47]. Finally, we use a self-administered paper questionnaire in the pre-test. The presentation of the French-language questionnaire for online use will need to be tested, especially as the tool may be incorporated in the future into patient-oriented platforms on smartphones or tablets [48]. 

## 5. Conclusions

The financial burden faced by PLC is increasing but remains poorly understood and insufficiently addressed. The French language P-SAFE questionnaire is a new tool that has the potential to help detect problems, develop person-centred care plans, and fill an important gap in whole-person care. The P-SAFE represents an opportunity to reduce the number of PLC falling through the cracks due to financial hardship and appears as a useful support in the development of cancer care plans. The translation and cultural adaptation process described in this paper provides valuable insight into how French-speaking PLC in Quebec understand the questions and response options, and results justify proceeding with further validation studies. The next step will be to implement the P-SAFE questionnaire, promote and sustain its use in practice settings, and ultimately evaluate its contribution to effective cancer care.

## Figures and Tables

**Figure 1 healthcare-08-00366-f001:**
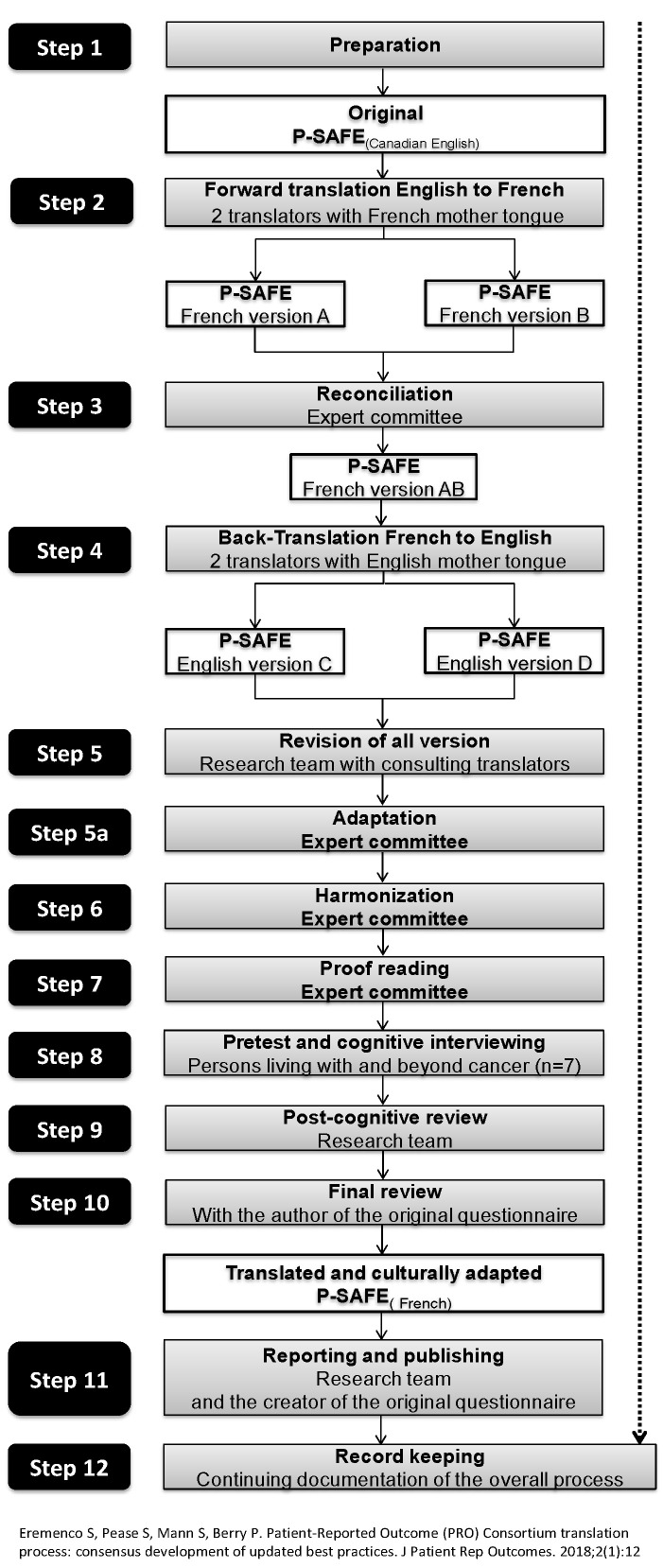
Translation and cultural adaptation flow diagram (adapted from Erememco et al. 2018) [39].

**Table 1 healthcare-08-00366-t001:** Summary of expert committee comments made during the translation and cultural adaptation processes.

Question Number	Comment	Issue	Decision
All	Expression: “Your cancer”	Appropriateness for people living with and beyond cancer	Replace “your cancer” by “a cancer” or just “cancer”
All	Terminology specific to the clinical context	Semantic equivalence	Verify the accuracy and adapt where needed
All	Passages using different phrasing to communicate the same meaning	Conceptual equivalence	Reconcile by retaining phrasings closer to those used in the original version
Q1	Terminology specific to insurance	Item equivalence	Verify the accuracy and adapt as needed
Q2	Terminology specific to insurance for life-threatening disease	Item equivalence	Verify the accuracy and adapt as needed
Q4	Uneven phrasing in subquestions 4a to 4j	Semantic equivalence and uniformity	Uniformize the phrasing
Q9	Negative phrasing: “lesser value housing”/“maison de valeur moindre”	Appropriateness for persons living with and beyond cancer	Replace by a positive phrasing: “more affordable housing”/“maison plus abordable”
Q12 and Q13	Designation of the participant in a column of the table: “patient”	Appropriateness for persons living with and beyond cancer	Replace “patient” by “you”
Q15	Choice of income categories	Comparability with national survey instruments	Harmonize with Statistics Canada categories
N.A.	Maladapted coping strategies remain unaddressed	Clinical relevance	Add a question: Have you or any of your caregivers taken any of the following actions for financial reasons? (a) Skip an appointment with your physician (b) Refuse a treatment (c) Postpone filling a prescription (d) Skip a dose of a prescribed drug (e) Cut pills in half (f) None of the above

**Table 2 healthcare-08-00366-t002:** Participant characteristics (*n* = 7).

Characteristics	*n*	(%)
Gender		
Men	4	0.57
Women	3	0.43
Education level		
University	2	0.29
College	4	0.57
High school	1	0.14
Tumour site		
Colorectal	3	0.43
Lung	2	0.29
Breast	1	0.14
Prostate	1	0.14
With metastases	2 *	0.29
Age group		
50–59	4	0.57
60–69	2	0.29
70 and above	1	0.14

Legend: * Metastases: bone; spine and stomach.

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
