# Peer review of "Translation and Cultural Adaptation of the Patient Self-Administered Financial Effects (P-SAFE) Questionnaire to Assess the Financial Burden of Cancer in French-Speaking Patients"

_healthcare, 2020, doi:10.3390/healthcare8040366_

Round 1

Reviewer 1 Report

Dear Authors,

Your paper entitled “Translation and cultural adaptation of the Patient Self-Administered Financial Effects (P-SAFE) questionnaire to assess the financial burden of cancer in French-speaking patients” provides a useful tool for assessing the financial hardship of people living with and beyond a cancer diagnosis. Financial burden might cause distress in cancer patients and might delay or forgo needed medical care, thus jeopardizing benefits of treatments.

While this is a valuable contribution to the field, I have some suggestions, listed below, to improve the impact of the results and their interpretation.

Comments:

1)      It would be worth to provide the reader with the translated French version of the P-SAFE questionnaire as supplementary material. This tool can be used as it is in the Quebec province or can be further adapted, following the same methodology, to other French speaking contexts. 

2)      The authors should provide some comments on the future use of the French version of the P-SAFE questionnaire in Quebec. What are the expected results of its application compared to the experience done with the corresponding English version in the rest of the country?

3)      In the Discussion section-Clinical and research implications, you mention the possibility to evaluate the utilization of the questionnaire on mobile electronic devices. It is not clear if this future development concerns both the English as well as the French version, or there are previous experiences with the English version. Please clarify adding a comment in the discussion.

Author Response

Response to Reviewer 1 comments:

Please note that as a result of several changes in the text and the addition of two new references, the line numbers, in-text citation numbers and references have changed. The conclusion has been modified to reduce the risk of self-citation (pointed out by the Editorial Assistant) and more clearly reiterate the paper's main contribution.

We thank Reviewer 1 for your valuable comments on the paper. We hope that the following responses adequately address the issues you raise.

Point 1:     It would be worth to provide the reader with the translated French version of the P-SAFE questionnaire as supplementary material. This tool can be used as it is in the Quebec province or can be further adapted, following the same methodology, to other French speaking contexts. 

Response 1: The developers of the original English-language P-SAFE questionnaire at McMaster University (Ontario, Canada) ask that researchers wishing to use the questionnaire contact them directly. Christopher Longo, co-author of the present study, leads the team at McMaster. They do not charge a fee but request that aggregate results of studies using the P-SAFE questionnaire be shared with their research team to provide a snapshot of the financial effects of cancer in different jurisdictions. We are happy to share the French questionnaire upon request with researchers who register a request with the McMaster research team at [email protected]. Please see https://psafe.mcmaster.ca/research-tools-request/ for more information.

      We include a note stating the above in the acknowledgements section of the article (lines 302-308).

Point 2;  The authors should provide some comments on the future use of the French version of the P-SAFE questionnaire in Quebec. What are the expected results of its application compared to the experience done with the corresponding English version in the rest of the country?

Response 2: We consider that including this information in the present paper would distract from its main focus: the translation and cultural adaptation of the tool. However, for the reviewers' information, The P-SAFE questionnaire is currently being used in a major study of the Quebec cancer network (Professor Dominique Tremblay, RCQ-2, funded by the Fonds de recherche Québec, grant 27193, and Oncopole, grant 265874) to evaluate the financial burden associated with the network organisation of cancer care services. Collaboration with Christopher Longo at McMaster University continues in order to refine the statistical analysis and contribute to assessment of survey validity, based on both English and French questionnaires.

Point 3: In the Discussion section-Clinical and research implications, you mention the possibility to evaluate the utilization of the questionnaire on mobile electronic devices. It is not clear if this future development concerns both the English as well as the French version, or there are previous experiences with the English version. Please clarify adding a comment in the discussion.

Response 3: The English P-SAFE questionnaire is currently available online in many Canadian provinces. While no comprehensive study of user-friendliness has yet been published, the online version appears to be well accepted by clinicians and PLC. The last section of 4.2 (line 277) has been amended to make our meaning clear: "The presentation of the French-language questionnaire for online use will need to be tested, especially as the tool may be incorporated in future into patient-oriented platforms on smart phones or tablets [47]."

Reviewer 2 Report

You (the authors) have made an important attempt to translate an instrument, and make it available to French speaking people living with cancer. However, there are some major and minor flaws in the execution of your work. 

  1. What is the purpose of the tool, to help the patients understand their own needs, communicate concerns for the provider? A better overview of the tool’s purpose, i.e. to communicate economic vulnerability needs to be included in the manuscript.
  2. The conclusion mentions person centered care plans, but does not provide information on how the tool has been used to improve the financial health of individuals who use it.
  3. Minor comments: In abstract, line 32 providing scale for the range would be useful for understanding impact of the number 5-7
  4. Line 61: The statement of PLC have a greater risk of recurrence (than people not diagnosed with cancer?) Does not make sense---a recurrence requires a baseline cancer diagnosis.

  5. The justification for the sample of individuals living with cancer needs to be made, why were they only ambulatory instead of a range of both inpatient and outpatient? And why the 4 diseases (or was it for the convenience of the investigators?) A stronger rationale for the population needs to be made, were all of the participants at the same point in treatment (line 152—completed first cycle of treatment). The number of participants (7) seems quite small for the validation study.

  6. The age group demographics are focused on older adults but in the preamble the authors mention that those under 65 have the most financial instability with cancer. No one in the group was younger than 50. 

  7. What generated the criteria of having completed one cycle of treatment? No justification for that is provided.

  8. The authors also mention that those with less education are at greater financial risk, but the sample does not include a large proportion of PLC who have a high school education or less. 

  9. The questions asked by the research assistant seem to be targeted to assessing the use of the instrument, not on the validity of it. The range on line 170 needs to be placed early in the manuscript, when the scale is first mentioned.

  10.  

    Was the coding independently confirmed by the two coders, line 173

Author Response

Response to Reviewer 2 Comments:

Please note that as a result of several changes in the text and the addition of two new references, the line numbers, in-text citation numbers and references have changed. The conclusion has been modified to reduce the risk of self-citation (mentioned by the Editorial Assistant) and more clearly reiterate the paper's main contribution.

Preliminary author note to Reviewer 2:

Thank you very much for your careful reading and insightful comments on our paper. Before addressing each of your points, we would first like to address an issue that appears to be at root of a number of the comments: these reflect an expectation of testing aspects of questionnaire validity (i.e. functional, psychometric) that are beyond the scope and ambition of the present study, which addresses only the content validity and provides a model process to assure clarity and cultural adaptation. We are concerned that this objective may not be sufficiently clear in the introduction to the article.

We have added the following after line 104:

This initial testing phase is a necessary but not sufficient condition for ascertaining the overall validity of a questionnaire. This initial testing phase is necessary to provide insight into how respondents understand and interpret the items and instructions on the translated questionnaire. The process aims to ensure consistency in the content and face validity between the English and French versions of the P-SAFE (Beaton, 2000; p. 3189). Content validity rests on qualitative, not quantitative, evaluation (Vallerand, 1989; p. 670).

Point 1: What is the purpose of the tool, to help the patients understand their own needs, communicate concerns for the provider? A better overview of the tool’s purpose, i.e. to communicate economic vulnerability needs to be included in the manuscript.

Response 1. The P-SAFE questionnaire provides a means of quantifying and understanding the economic cost burden on patients of cancer treatment. Studies using the questionnaire can fill information gaps to inform government policy around coverage of medical and non-medical goods and services; employment policy; health services organization; care team support and information for patients. We recognize that the sentence in lines 92-93 may emphasize just the healthcare professional role in addressing financial questions.

We have reworked the paragraph (lines 89-95) to read: "The financial burden may vary between PLC according to type of cancer and treatment, as well as factors such as complementary private insurance coverage for some cancer drugs; family support; employment situation; distance from cancer centre, etc. Better understanding of the financial burden of cancer is needed to improve decision-making at multiple levels, from government policy around coverage, to health services organization, employment practices, treatment choices and more [34]. Interestingly, a recent study found that PLC did not mention financial distress to their healthcare professionals [35]. In publicly funded health systems, many people, including healthcare providers, believe expenses for serious disease are covered, and may be less aware and less equipped to address financial questions [8, 36]. The need to raise awareness of and attention to the growing financial burden of cancer motivated the effort, described in this paper, to translate and adapt the P-SAFE questionnaire for French speaking PLC in Quebec."

Point 2: The conclusion mentions person centered care plans, but does not provide information on how the tool has been used to improve the financial health of individuals who use it.

Response 2: To our knowledge, there has not yet been study of improvements based on results obtained from the P-SAFE questionnaire. The conclusion refers to the "potential" (line 283) of the tool to develop person-centred care plans and design whole-person care. This would entail efforts at multiple levels to relieve the financial burden.

Point 3: Minor comments: In abstract, line 32 providing scale for the range would be useful for understanding impact of the number 5-7

Response 3: Thank you for the comment. We propose rewording the sentence (line 32-33): "The mean score for clarity of questions was 6.4 out of a possible 7 (totally clear)".

Point 4: Line 61: The statement of PLC have a greater risk of recurrence (than people not diagnosed with cancer?) Does not make sense---a recurrence requires a baseline cancer diagnosis.

Response 4: It is the heavier financial burden that is associated with greater risk of recurrence. Line 63: "PLC with a heavier financial burden...". We have left the sentence as is.

Point 5: The justification for the sample of individuals living with cancer needs to be made, why were they only ambulatory instead of a range of both inpatient and outpatient? And why the 4 diseases (or was it for the convenience of the investigators?) A stronger rationale for the population needs to be made, were all of the participants at the same point in treatment (line 152—completed first cycle of treatment). The number of participants (7) seems quite small for the validation study.

Response 5: The rationale for sample criteria was as follows: Outpatient (ambulatory) treatment entails a greater risk of cost burden, from travel and parking, to lost income, to home help and additional medications to counter side effects, etc.; having completed a first cycle of treatment assured that patients had some exposure to such costs (we did not consider it essential that they be at the same point in treatment); the four cancers are those most commonly diagnosed. In terms of sample size, seven participants are considered sufficient for validating the translation of tools (Poder). A larger sample would be required for psychometric and other forms of validation.

Point 6: The age group demographics are focused on older adults but in the preamble the authors mention that those under 65 have the most financial instability with cancer. No one in the group was younger than 50. 

Response 6. The question is pertinent as younger patients face more important financial burdens. However, the objective in the present study was to assess the clarity of the questionnaire and not the financial burden itself.

Point 7: What generated the criteria of having completed one cycle of treatment? No justification for that is provided.

Response 7. Thank you. The rationale for selecting this group was to assure respondents had been exposed to costs related to cancer and treatment.

We have added the following sentence (at line 175): "Ambulatory patients having completed at least a first cycle of treatment were selected as they were sure to have been exposed to cost burdens."

Point 8: The authors also mention that those with less education are at greater financial risk, but the sample does not include a large proportion of PLC who have a high school education or less. 

Response 8: We consider that the inclusion of participants with a wide range of education levels would be necessary in a study of questionnaire validity. We also recognize that literacy presents an important impediment to participation in research more generally and that efforts are needed to support the participation of people with low literacy. However, with the modest aim of assessing the clarity of the French translation of the questionnaire, we considered that the educational range from high school to advanced degree was acceptable.

Point 9: The questions asked by the research assistant seem to be targeted to assessing the use of the instrument, not on the validity of it. The range on line 170 needs to be placed early in the manuscript, when the scale is first mentioned.

Response 9: The debriefing questions were intended to assess the functional equivalency of the French questionnaire to the original English version. No part of the present study sought to evaluate the validity of the questionnaire. Questions during the debriefing were designed to assess clarity and cultural adaptation. The time it took participants to complete the questionnaire (line 181) provides an additional indication of clarity.

We have moved up the scale for clarity from line 193 to line 182-83)

Point 10: Was the coding independently confirmed by the two coders, line 173

Response 10: Yes, the two co-authors coded results of the think-aloud and debrief sessions independently, achieving consensus, with some input from the expert committee, on interpretations.

We have added (line 197): "and achieving consensus on interpretations".

Round 2

Reviewer 2 Report

The authors have responded completely to the first round of suggestions, my only final parting comment is that there is no mention of the need to further test for reliability and instead in the conclusion, the authors suggest that it should be moved immediately into clinical practice. In the response to my concerns, the authors responded:

"This initial testing phase is a necessary but not sufficient condition for ascertaining the overall validity of a questionnaire. This initial testing phase is necessary to provide insight into how respondents understand and interpret the items and instructions on the translated questionnaire. The process aims to ensure consistency in the content and face validity between the English and French versions of the P-SAFE (Beaton, 2000; p. 3189). Content validity rests on qualitative, not quantitative, evaluation (Vallerand, 1989; p. 670)" 

Consequently, I am curious that the revised conclusion stated: "The next step will be to 264 implement the P-SAFE questionnaire, promote and sustain its use in practice settings, and ultimately 265 evaluate its contribution to effective cancer care." 

While the instrument and its' focus is important for people living with and beyond cancer, French speaking individuals who the tool is designed for, deserve the benefit of a full assessment of the instrument. 

This is an important contribution to the literature and I believe that it should be published, but recommend clarity when discussing the next recommended step. 

Thank you.